# A genetically encoded sensor for visualizing leukotriene B4 gradients in vivo

**Szimonetta Xénia Tamás** [1,2,3], **Benoit Thomas Roux** [1,3], **Boldizsár Vámosi** [1], **Fabian Gregor Dehne** [1,3], **Anna Török** [1,3], **László Fazekas** [1,2,3] & **Balázs Enyedi** [1,2,3] ✉

Leukotriene B4 ($LTB_4$) is a potent lipid chemoattractant driving inflammatory responses during host defense, allergy, autoimmune and metabolic diseases. Gradients of $LTB_4$ orchestrate leukocyte recruitment and swarming to sites of tissue damage and infection. How $LTB_4$ gradients form and spread in live tissues to regulate these processes remains largely elusive due to the lack of suitable tools for monitoring $LTB_4$ levels in vivo. Here, we develop GEM-$LTB_4$, a genetically encoded green fluorescent $LTB_4$ biosensor based on the human G-protein-coupled receptor BLT1. GEM-$LTB_4$ shows high sensitivity, specificity and a robust fluorescence increase in response to $LTB_4$ without affecting downstream signaling pathways. We use GEM-$LTB_4$ to measure ex vivo $LTB_4$ production of murine neutrophils. Transgenic expression of GEM-$LTB_4$ in zebrafish allows the real-time visualization of both exogenously applied and endogenously produced $LTB_4$ gradients. GEM-$LTB_4$ thus serves as a broadly applicable tool for analyzing $LTB_4$ dynamics in various experimental systems and model organisms.

Gradients of chemoattractants guide leukocyte migration during immune surveillance. Neutrophils as first responders during inflammation are remarkably efficient in interpreting directional cues which recruit them through the vessel wall to sites of tissue damage and infection[1,2]. Their migration is guided by a vast array of chemically diverse chemoattractants including bacterial peptides, complement fragments, lipid mediators and various chemokines[3]. Gradients of primary chemoattractants such as N-formylated peptides (fMLP) and complement 5a (C5a) attract the first neutrophils to inflammation sites, where their activation contributes to a cascade of secondary chemoattractant production[4]. Among these chemoattractants, leukotriene B4 ($LTB_4$) acts as a central signal relay molecule, self-amplifying its production to increase the detection range and enhance the robustness and persistence of leukocyte migration during host defense[5].

$LTB_4$ is synthesized on the nuclear envelope by the sequential action of 5-lipoxygenase (5-LOX) and leukotriene A4 hydrolase from arachidonic acid, which is released from phospholipids by cytosolic phospholipase A2 (cPLA2)[6]. Calcium transients and mechanical stretch on the nuclear membrane jointly contribute to the activation of cPLA2 and 5-LOX by translocating them to the nuclear membrane[7]. $LTB_4$ is then packaged along with its synthesizing enzymes into vesicles, which bud off from the nuclear envelope, and are ultimately secreted as exosomes from activated neutrophils, macrophages and dendritic cells[8,9]. Although its secretion and distribution have not been directly visualized, $LTB_4$ has been proposed to form local and long-range chemoattractant gradients to drive directional migration through its G protein-coupled receptor, BLT1[5,10,11]. $LTB_4$ has also been established as a main driver of neutrophil swarming. This emergent behavior driving collective neutrophil migration to large targets is dependent on the combination of transcellular $LTB_4$ production and an $LTB_4$-BLT1 axis dependent self-amplification[12–15]. A major obstacle to gain further insight into the regulation of these processes has been the lack of tools to directly measure the real-time production of $LTB_4$.

[1]Department of Physiology, Semmelweis University, Faculty of Medicine, Tűzoltó utca 37-47, H-1094 Budapest, Hungary. [2]MTA-SE Lendület Tissue Damage Research Group, Hungarian Academy of Sciences and Semmelweis University, H-1094 Budapest, Hungary. [3]HCEMM-SE Inflammatory Signaling Research Group, Department of Physiology, Semmelweis University, H-1094 Budapest, Hungary. ✉e-mail: enyedi.balazs@med.semmelweis-univ.hu

Measuring $LTB_4$ and other chemoattractant levels with high spatiotemporal resolution requires methods beyond standard biochemical assays. Previous approaches to assess the tissue distribution of chemoattractants and chemokines have been based on immunofluorescence assays or transgenic labeling of endogenous chemokines with fluorescent proteins[16–19]. Alternatively, live measurements of receptor internalization have been used to approximate endogenous chemokine gradients[20]. However, this method is insufficient to follow the events in real-time. In addition, it is not amenable to ligands inducing weak receptor internalization such as $LTB_4$[21]. To overcome these limitations of detection, we took an approach inspired by recent advances in the neuroscience field, which have led to the development of GPCR-based fluorescent biosensors for the live imaging of neurotransmitters[22–25].

Here, we develop a genetically encoded fluorescent reporter for the direct, rapid and sensitive measurement of extracellular $LTB_4$ levels. The sensor called GEM-$LTB_4$ is structurally based on the human BLT1 receptor with a circularly permutated EGFP (cpEGFP) inserted into the third intracellular loop of the GPCR. GEM-$LTB_4$ shows a robust fluorescence response that allows us to visualize not only exogenously applied but also endogenous gradients of $LTB_4$ both in vitro and in vivo.

## Results

### Development and characterization of GEM-$LTB_4$ in HEK293A cells

To develop an $LTB_4$ sensor, we inserted a cpEGFP module with linkers into the third intracellular loop of the high-affinity $LTB_4$ receptor, BLT1 between R212 and F213 (Fig. 1a, b). Linker sequences on the N- and C-termini of cpEGFP were designed based on the previously published GPCR-based dLight sensors[22]. In our prototype sensors, combinations of long and short linkers were tested (Supplementary Fig. 1a). When expressed in HEK293A cells, these sensors showed a modest fluorescence response upon stimulation with $LTB_4$ (Supplementary Fig. 1b). Based on the linker sequences of the sensor exhibiting the largest response (linker NL-CS: $\Delta F/F_0 = 35.9 \pm 2.5\%$, mean ± SEM), we performed a second round of screening by linker length optimization. This screening step identified that the four amino acid long NHDQ linker on the C-terminal end of cpEGFP gives rise to sensors with enhanced dynamic range (Supplementary Fig. 1c). In parallel measurements we evaluated the plasma membrane localization of the sensors exhibiting the highest $\Delta F/F_0$ and identified the version (N4-C4) which shows the best combination of dynamic range ($\Delta F/F_0 = 103 \pm 1\%$, mean ± SEM) and membrane targeting (Supplementary Fig. 1d, e). To further enhance its membrane trafficking[23], we attached the IgK leader sequence to our final sensor which we called GEM-$LTB_4$ (Fig. 1c). Confocal imaging of GEM-$LTB_4$-expressing cells revealed that the brightness of the sensor in $LTB_4$-bound state was ~7-fold lower than that of a GFP-tagged BLT1 (Supplementary Fig. 1f).

To create a control sensor variant (GEM-$LTB_4$mut) we introduced a point mutation corresponding to R156A in BLT1, which abolishes $LTB_4$ binding[26]. While this mutant sensor localizes well to the plasma membrane, it does not change its fluorescence upon stimulation with $LTB_4$ (Fig. 1c). For signal normalization we co-expressed the plasma membrane-targeted red fluorescent mKate2 protein beside GEM-$LTB_4$ using the self-cleaving viral P2A peptide[27]. This resulted in significantly better sensor expression levels than with direct mKate2 fusion of the sensor (unpaired t-test, $p = 0.003331$, Supplementary Fig. 1g, h).

Spectral characterization of GEM-$LTB_4$ showed an $LTB_4$-dependent excitation peak between 450-500 nm with an isosbestic point at 425 nm (Fig. 1d). The apparent affinity of GEM-$LTB_4$ in HEK293A cells is in the low nanomolar concentration range with an $EC_{50}$ of ~19.8 nM (Fig. 1e, f). $LTB_4$-induced fluorescence increase was stable over 45 min, and was reversible by BLT1 inhibitors or by removal of $LTB_4$ from the media (Fig. 1g, Supplementary Fig. 2a–d, Supplementary Video 1). We

measured activation kinetics of GEM-$LTB_4$ using rapid local perfusion combined with high-speed imaging and calculated an average activation time constant of $770 \pm 37$ ms (Fig. 1h). Next, we applied a series of eicosanoid ligands to reveal that the specificity profile of GEM-$LTB_4$ is similar to its parent receptor, BLT1[28]. Of the compounds tested, as expected, only the BLT1 agonists $LTB_4$ and 20-OH-$LTB_4$ induced significant changes in fluorescence (One way ANOVA, $p = 2.0 \times 10^{-181}$, Fig. 1i). To evaluate whether GEM-$LTB_4$ is sensitive to changes in intracellular pH, we measured its response in nigericin and monensin-treated cells using intracellular buffers set to a range of pH values. As expected from a cpEGFP-based biosensor, when compared to the basal value at pH 7.4, GEM-$LTB_4$ fluorescence decreased or increased between −66% to +38% in the range of pH values between 6.2–8.6 (Supplementary Fig. 3a). At the same time, the sensor retained its responsiveness to $LTB_4$ across the range of measured pH values (Supplementary Fig. 3b).

We then assessed the coupling of GEM-$LTB_4$ to downstream cellular signaling pathways. While $LTB_4$ induces intracellular $Ca^{2+}$ signals in BLT1-expressing HEK293A cells due to known G-protein coupling[28], no $Ca^{2+}$ transients were seen in GEM-$LTB_4$ expressing cells after stimulation (Supplementary Fig. 4a, b, Supplementary Video 1). We also quantified the plasma membrane recruitment of β-arrestin-2, which was only significant in $LTB_4$ stimulated BLT1 expressing cells (Pairwise t-test, $p = 3.0 \times 10^{-6}$, Supplementary Fig. 4c, d). GEM-$LTB_4$ did not recruit β-arrestin-2 despite prolonged incubation with $LTB_4$, which is consistent with the lack of internalization and the stable fluorescence of the sensor seen in cells during long exposure to $LTB_4$ (Supplementary Fig. 4a, b). These results indicate that GEM-$LTB_4$ has a minimal potential to interfere with endogenous signal transduction pathways and allows long-term direct measurements of $LTB_4$ levels without desensitization.

### Imaging $LTB_4$ release from murine neutrophils with GEM-$LTB_4$

We next evaluated the performance of GEM-$LTB_4$ in detecting neutrophil-derived endogenous $LTB_4$ production. Isolated murine neutrophils were stimulated with fMLP and seeded on top of stable GEM-$LTB_4$-expressing cells. Applying neutrophils prestimulated for 30 min with fMLP to cells expressing the sensor gave rise to an instantaneous fluorescence signal increase (Supplementary Fig. 5a–d) indicating the presence of $LTB_4$ in the stimulating solution. This was verified by ELISA, showing $LTB_4$ secretion by neutrophils similar to previously reported values[5] (Fig. 2a). Application of fMLP or non-stimulated neutrophils alone did not change GEM-$LTB_4$ fluorescence (Supplementary Fig. 5b–d). To measure real-time kinetics of $LTB_4$ production we stimulated neutrophils with fMLP while they were sedimenting on top of sensor-expressing cells. GEM-$LTB_4$ fluorescence increased in areas densely covered by neutrophils (2000 cells/0.1 $mm^2$)- 5 min after fMLP stimulation (Fig. 2b, c). To confirm that the observed signals were due to $LTB_4$ release, we verified that fMLP-stimulated neutrophils do not elicit responses in GEM-$LTB_4$mut-expressing cells (Fig. 2b, c). $LTB_4$ production starts in foci and spreads over larger fields (Supplementary Video 2). Automatic quantification identified significant GEM-$LTB_4$ signal increase in $38 \pm 5.4\%$ (mean ± SEM) of the surface area imaged with the sensor cells (Fig. 2d, Unpaired t-test, $p = 0.001159$). Next, we measured $LTB_4$ release in areas of lower neutrophil density to potentially capture signals from individual cells. In this experimental setup after fMLP stimulation, we could detect GEM-$LTB_4$ signals radially emanating from areas where individual neutrophils were residing (Fig. 2e, f, Supplementary Video 2). Supporting that these individual neutrophils are actively producing chemoattractants, pseudopod formation in surrounding neutrophils could be detected towards the producing cells (Supplementary Video 2). These results demonstrate that the sensitivity of GEM-$LTB_4$ is sufficient to detect endogenous $LTB_4$ secretion from neutrophils ex vivo.

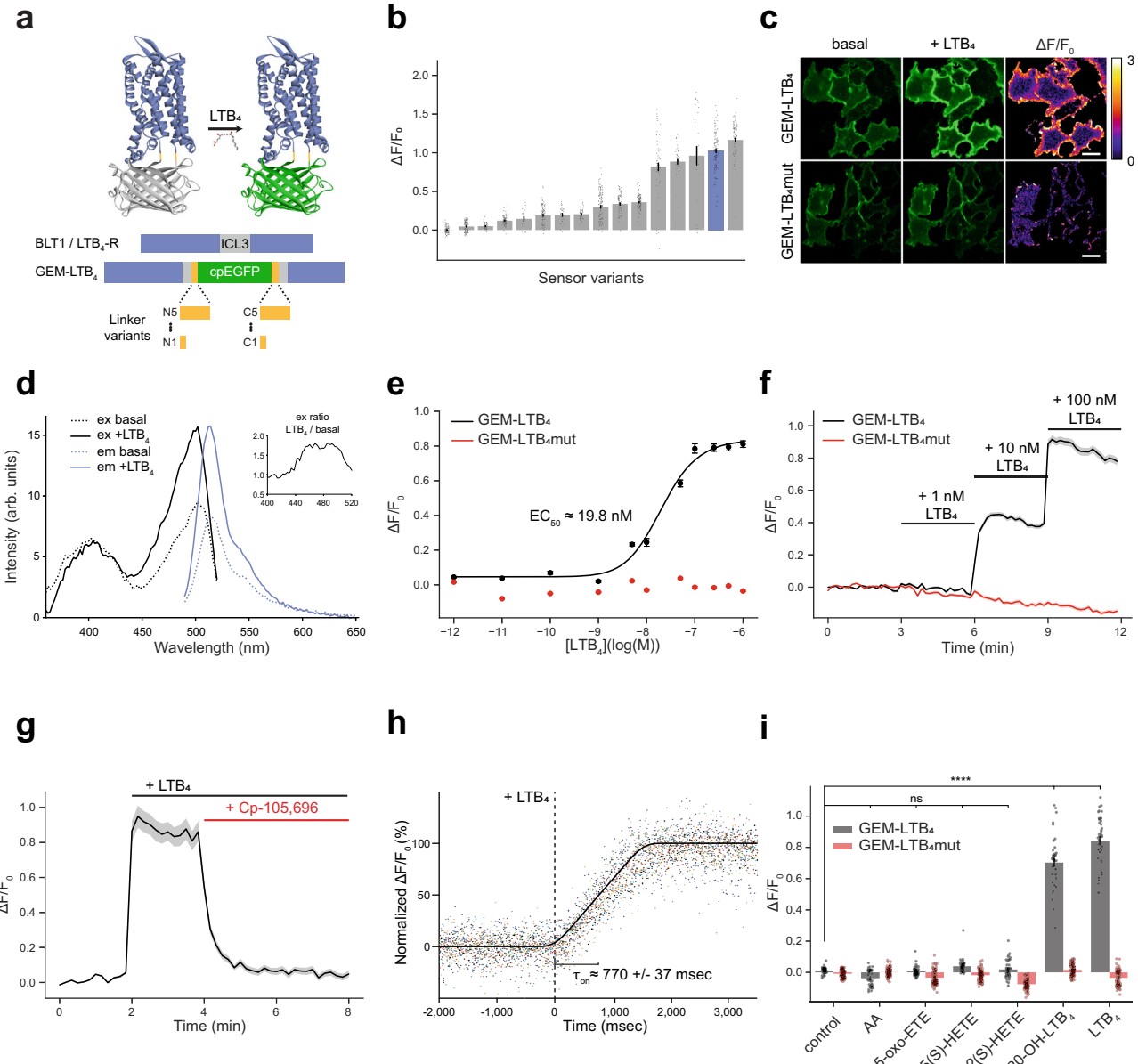

**Fig. 1 | Development and characterization of the GEM-LTB4 sensor in HEK293A cells. a** Schematic diagram of LTB4-sensor design showing fluorescence increase upon ligand binding. The sensor consists of cpEGFP inserted with linkers into the 3rd intracellular loop (ICL3) of BLT1. **b** Summary of $\Delta F/F_0$ fluorescence responses in all LTB4-sensor variants tested in this study. Data shown as mean ± SEM for $n = 106$, 134, 61, 90, 17, 105, 30, 62, 135, 98, 139, 62, 47, 26, 97 and 86 cells from 3 independent experiments, respectively. GEM-LTB4 is shown in blue. **c** Representative GEM-LTB4 and GEM-LTB4mut confocal fluorescence and corresponding $\Delta F/F_0$ ratio images in HEK293A cells before and after 100 nM LTB4 stimulation. Scale bars, 25 μm. **d** Excitation and emission spectra of GEM-LTB4 in the absence (dotted lines) and presence (continuous lines) of 100 nM LTB4. Insert shows ratio of excitation spectra. Each trace is the average of $n = 3$ independent experiments (arb. units=arbitrary units). **e** Dose-response measurements of GEM-LTB4 and GEM-LTB4mut, with the corresponding $EC_{50}$ value. Data shown as mean ± SEM for $n = 66$ and 218 cells per condition, respectively, from 3 independent experiments. $EC_{50}$

value was obtained by fitting the data to a four-parameter log-logistic function. **f** Average $\Delta F/F_0$ responses of GEM-LTB4 and GEM-LTB4mut to sequentially added increasing doses of LTB4. Data shown as mean ± SEM for $n = 113$ and 181 cells, respectively, from 3 independent experiments. **g** GEM-LTB4 response to 100 nM LTB4 stimulation followed by treatment with 1 μM of the BLT1 inhibitor CP-105,696. Data shown as mean ± SEM for $n = 85$ cells from 3 independent experiments. **h** Kinetic analysis from high-speed acquisition of GEM-LTB4 fluorescence in HEK293A cells during 100 nM LTB4 stimulation. All measured normalized data points and the average fitted curve are shown from $n = 13$ cells from 7 independent experiments. **i** Maximal $\Delta F/F_0$ responses of GEM-LTB4 and GEM-LTB4mut to 100 nM of the indicated eicosanoid compounds. Data shown as mean ± SEM for $n = 45$ and 88 cells, respectively, from 3 independent experiments. Statistical analysis were performed with One-way ANOVA ($F = 761.3$, $p = 2.0 \times 10^{-181}$) with Dunnett's correction (20-OH-LTB4 and LTB4 are different from control for GEM-LTB4). Source data are provided as a Source Data file.

## Visualizing LTB4 levels in zebrafish larvae with GEM-LTB4

In order to demonstrate that GEM-LTB4 is suitable for in vivo detection of LTB4, we created transgenic zebrafish lines expressing the sensor under the control of suprabasal (*krt4*) and basal (*krt19*) epidermal skin layer-specific keratin promoters. We measured the penetration of exogenous LTB4 into the tail fin of amputated zebrafish larvae kept in isotonic embryo media (see Methods for details), which prevents early leukocyte recruitment and wound closure[29, 30], thereby allowing better penetration of exogenously applied substances through the open wound. By confocal imaging, GEM-LTB4 showed plasma membrane localization and an LTB4-dependent fluorescence signal increase close to 100% in both epidermal layers (Fig. 3a, b), which is similar to the

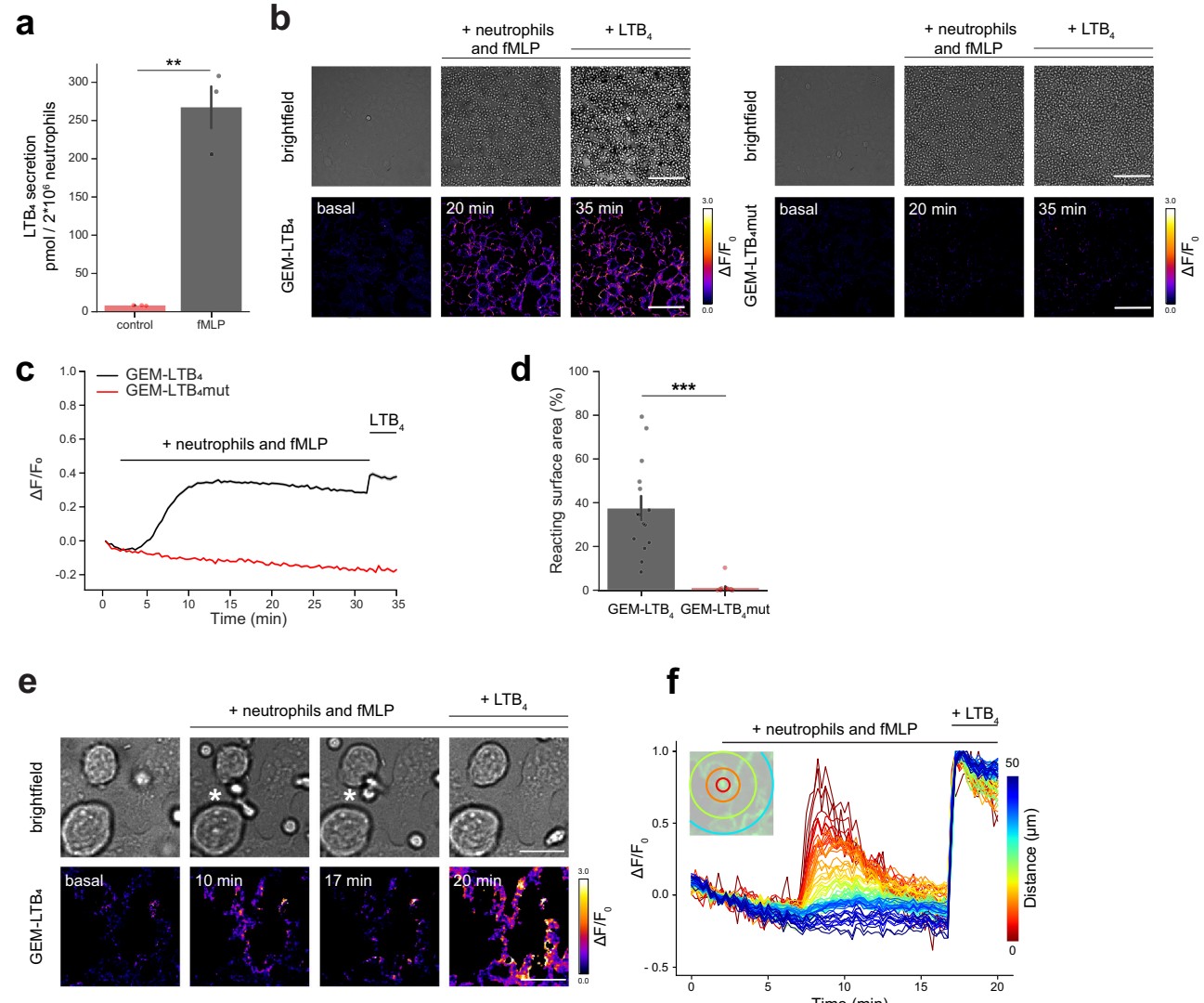

**Fig. 2 | Real-time measurements of LTB$_4$ release from neutrophils with GEM-LTB$_4$.** **a** ELISA measurement of LTB$_4$ secretion by murine neutrophils stimulated with 2 μM fMLP for 28 min. Data shown as mean ± SEM for 3 independent experiments, two-tailed unpaired $t$-test, $^{**}P = 0.0012$. **b** Brightfield microscopy and corresponding ΔF/F$_0$ ratio images of murine neutrophils seeded over HEK293A cells expressing GEM-LTB$_4$ (*left*) and GEM-LTB$_4$mut (*right*). Scale bars, 100 μm. Representative images were taken before neutrophil addition (basal), after adding neutrophils and stimulating with 2 μM fMLP (20 min) and followed by 100 nM LTB$_4$ stimulation (35 min). **c** Average traces of ΔF/F$_0$ responses in GEM-LTB$_4$ and GEM-LTB$_4$mut expressing cells shown in **b**. Data are presented as mean ± SEM for $n = 600$ and 465 cells respectively from 3 independent experiments. **d** Relative surface area of GEM-LTB$_4$ expressing cells shown in **b**, reacting with over 50% increase in

normalized ΔF/F$_0$ as a response to $2 \times 10^6$ neutrophils stimulated with 2 μM fMLP in a 1 cm$^2$ chamber. Data shown as mean ± SEM for $n = 15$-15 fields of view from 3-3 independent experiments with two-tailed unpaired $t$-test, $^{***}P = 0.001159$. **e** Brightfield microscopy and corresponding ΔF/F$_0$ ratio images of GEM-LTB$_4$ expressing HEK293A with neutrophils seeded over them at a ~1/10 density compared to **b**. Representative ΔF/F$_0$ images taken before neutrophil addition (basal) and at two time points after stimulation with 2 μM fMLP followed by 100 nM LTB$_4$. White asterisk (*) refers to the center of the analysis shown in **f**. Scale bars, 20 μm. **f** Representative spatiotemporal traces of pixelwise ΔF/F$_0$ GEM-LTB$_4$ values from **e**. The spatial origo is the center of the marked neutrophil (*) also shown in the inlay image. Similar results were obtained in $n = 3$ independent experiments. Source data are provided as a Source Data file.

ΔF/F$_0$ response measured in HEK293A cells. We then compared the spatiotemporal LTB$_4$ signal distribution in the tail fin of intact and amputated zebrafish larvae expressing the sensor in the suprabasal layer. After LTB$_4$ application, we measured limited GEM-LTB$_4$ signal in intact fins compared to the gradient seen in amputated larvae (Fig. 3a–d, Supplementary Video 3), suggesting that the intact surface epithelium acts as a barrier for LTB$_4$ penetration. In larvae expressing the control GEM-LTB$_4$mut sensor, LTB$_4$ did not alter the fluorescence (Fig. 3d). Corresponding with the visualized spatiotemporal LTB$_4$ distribution, exogenous LTB$_4$ triggered neutrophil migration in tail fin of amputated larvae with a time course matching the measured gradient (Fig. 3e).

To assess the ligand buffering potential of GEM-LTB$_4$, we further tested the effect of LTB$_4$ in a range of concentrations on leukocyte

migration in zebrafish. As previously shown, LTB$_4$ induces the dissemination of leukocytes from the caudal haematopoietic tissue (CHT) into the fins[31, 32]. The extent of leukocyte mobilization from the CHT is dose-dependent (Supplementary Fig. 6). Consistently with the previously proposed ligand-buffering capability of GPCR-based sensors[33], GEM-LTB$_4$ overexpression in the tail fin resulted in a decreased LTB$_4$ sensitivity in the low nanomolar range (30 nM) compared to GEM-LTB$_4$mut and control larvae (Supplementary Fig. 6).

## Measuring endogenous LTB$_4$ production in zebrafish larvae with GEM-LTB$_4$

LTB$_4$ regulates neutrophil swarming during tissue damage across species[12, 34, 35]. To capture endogenous LTB$_4$ release in zebrafish during

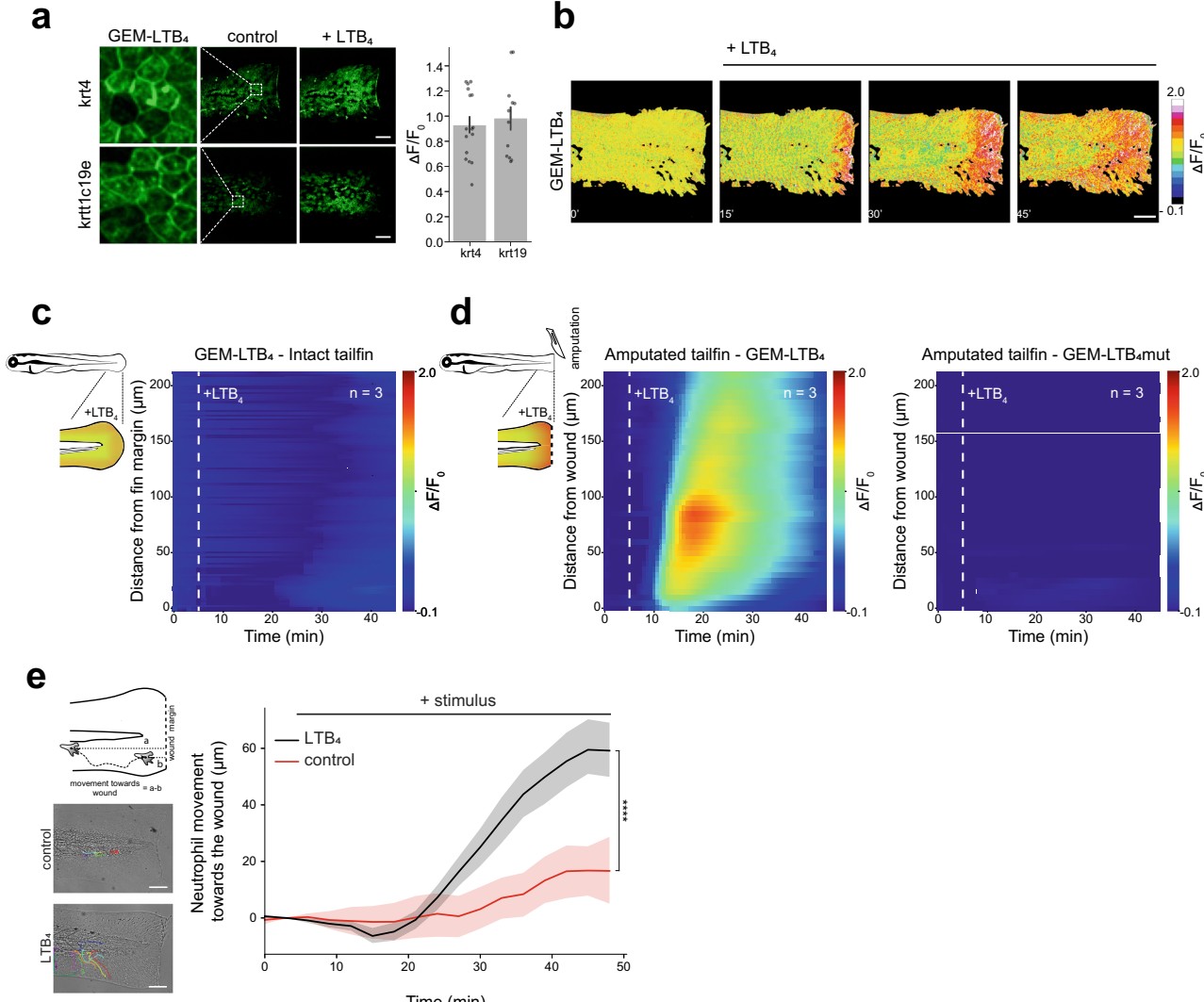

**Fig. 3 | Expression of GEM-LTB₄ in zebrafish and exogenous LTB₄ penetration detection.** **a** Representative confocal fluorescence imaging and quantification of $\Delta F/F_0$ responses of GEM-LTB₄ in *Tg(krt4:QF2 x QUAS:GEM-LTB₄)* (*top*) and *Tg(krt19:QF2 x QUAS:GEM-LTB₄)* (*bottom*) zebrafish larvae before and after 1 μM LTB₄ stimulation. Enlarged images show the cellular distribution of GEM-LTB₄ expression in the respective epithelial layers. Scale bars, 100 μm and data are presented as mean ± SEM for $n = 12$ and 16 cells from 3 and 4 independent fish, respectively. **b** Representative $\Delta F/F_0$ of time-lapse images of amputated zebrafish larvae *Tg(krt4:QF2 x QUAS:GEM-LTB₄)* after stimulation with 1 μM LTB₄. Scale bar, 100 μm. **c** Averaged spatiotemporal profile plot of GEM-LTB₄ $\Delta F/F_0$ responses after stimulation of intact tail fins with 1 μM LTB₄ in *Tg(krt4:QF2 x QUAS:GEM-LTB₄)* larvae. $n = 3$ larvae. **d** Averaged spatiotemporal profile plot of GEM-LTB₄ (*left*) and GEM-LTB₄mut (*right*) $\Delta F/F_0$ responses after stimulation of amputated tail fins with 1 μM LTB₄ in *Tg(krt4:QF2 x QUAS:GEM-LTB₄)* and *Tg(krt4:QF2 x QUAS:GEM-LTB₄mut)* larvae. The amputation and stimulation were performed under isotonic conditions (see Methods for details). $n = 3$ larvae. **e** Measurement of neutrophil movement triggered by control or LTB₄ towards amputational tail fin wounds imaged in *Tg(mpx:GFP)i114* zebrafish larvae by light transmission and fluorescence microscopy. *Top left*: scheme of neutrophil movement quantification towards the wound. *Left*: representative leukocyte tracks capturing all visible cell movements during imaging in control and 1 μM LTB₄ treated samples. *Right*: Time course of average neutrophil movement towards the wound shown in b, in control and in 1 μM LTB₄ stimulated larvae. Data are shown as mean ± SEM for $n = 25$ and 38 cells from 5 and 4 independent experiments, respectively, with a two-tailed unpaired t-test, $^{****}P = 1.6 \times 10^{-9}$ performed at the endpoint of the measurement. Source data are provided as a Source Data file.

sterile tissue injury, we measured GEM-LTB₄ signals in the basal epithelial cells of the tail fin. This is the cell layer closest to neutrophils and other leukocytes that are migrating towards the wound. We indeed detected LTB₄ production after sterile injury (Supplementary Video 4), however, only in rare instances which is entirely consistent with previous reports showing stochastic swarm development upon wounding in zebrafish[35,36]. To trigger endogenous LTB₄ release consistently from neutrophils, we used an established protocol which relies on recruiting leukocytes to a wound and activating their 5-LOX dependent LTB₄ production with the $Ca^{2+}$ ionophore A23187 at the same time[7,35] (Fig. 4a). We used arachidonic acid to recruit leukocytes to an open wound under isotonic conditions, which is a precursor readily

transformed to epithelial chemoattractants such as 5-oxoETE in cells around the wound margin[29]. Stimulating the larvae subsequently with A23187 resulted in the generation of real-time endogenous LTB₄ gradients emanating from the wound margin, where leukocytes could be detected by brightfield microscopy (Fig. 4b–d, Supplementary Video 5). Although the exact cellular source of LTB₄ was not determined, the presence of neutrophils and their ionophore-triggered $Ca^{2+}$ signal was confirmed in parallel experiments using a neutrophil-specific GCaMP7s-expressing transgenic line (Supplementary Video 5a). A change in GEM-LTB₄ fluorescence was not seen in larvae pretreated with the 5-LOX inhibitor zileuton (Fig. 4b–d, Supplementary Video 5b) or in the GEM-LTB₄mut expressing control larvae

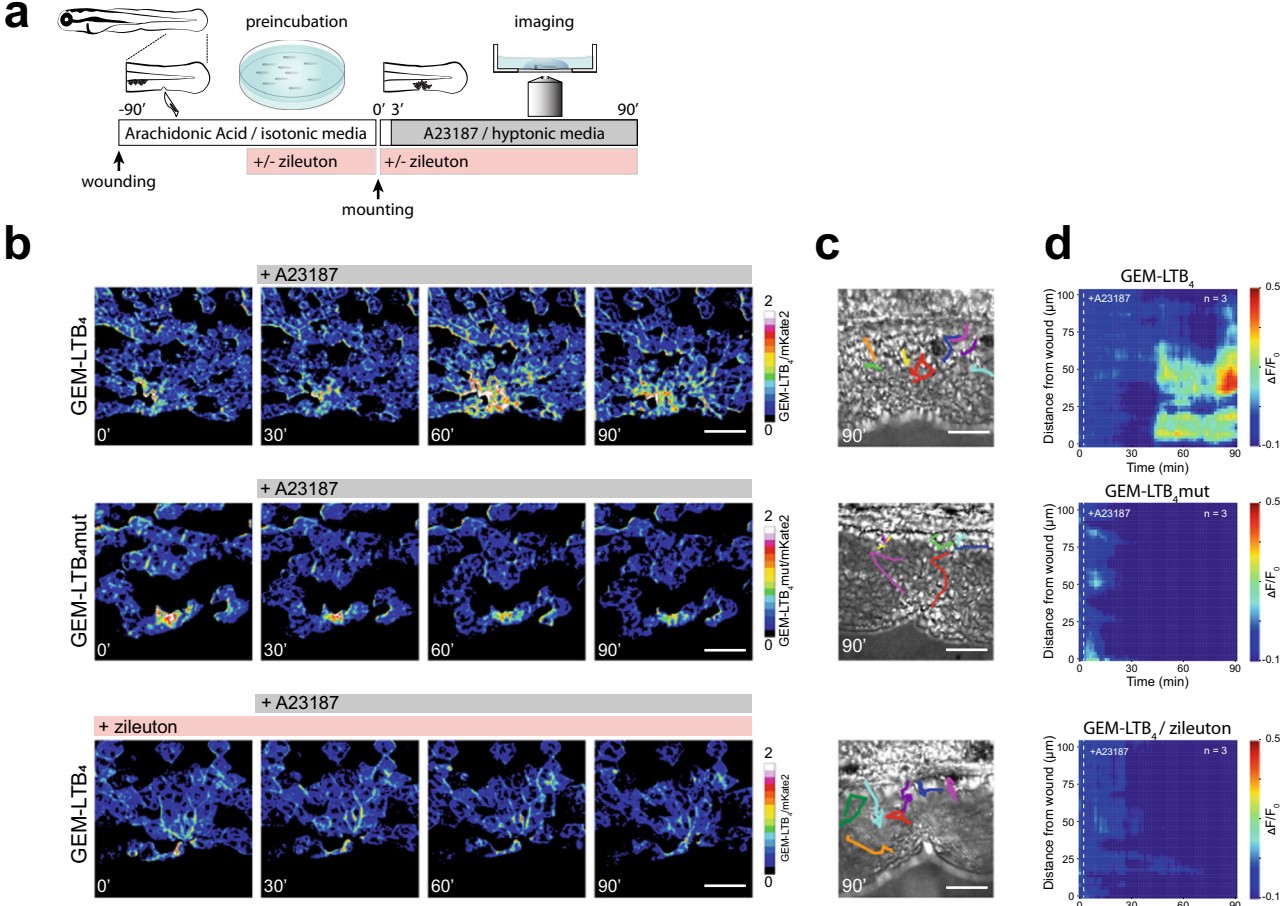

**Fig. 4 | Endogenous LTB₄ release measured by GEM-LTB₄ in zebrafish.**
**a** Schematics of experimental design for measuring endogenous LTB₄ release. Larvae were wounded and pre-incubated in isotonic E3 with 20 μM arachidonic acid for 90 min alone or in combination with 20 μM of the 5-lipoxygenase inhibitor zileuton, and then stimulated with 100 μM A23187 after mounting in hypotonic agarose. **b** Representative pseudo-color of green/red ratio of time-lapse images of zebrafish larvae *Tg(krt19:QF2 x QUAS:GEM-LTB₄)* (top and bottom) and *Tg(krt19:QF2 x QUAS:GEM-LTB₄mut)* (middle) in the basal epithelial cells. Scale bars, 50 μm. **c** Brightfield images of the corresponding wounded tail fins in b, at 90 min with manually traced leukocyte time lapse tracks shown in color. Scale bars, 50 μm. **d** Averaged spatiotemporal profile plots of corresponding experiments shown in **b**. *n* = 3 larvae each.

(Fig. 4b, d) nor did A23189 trigger responses in GEM-LTB₄ on its own (Supplementary Fig. 7a, b).

## Discussion

In summary, we developed and characterized a fluorescent biosensor for the live imaging of LTB₄ release. GEM-LTB₄ has the sensitivity, ligand specificity, photo-stability and kinetics suitable to measure physiologically relevant endogenous LTB₄ production. Importantly, GEM-LTB₄ does not activate downstream signaling pathways, nor does it internalize upon long-term stimulation, which are both required for the reliable measurement of extracellular LTB₄ dynamics. Indeed, GEM-LTB₄ allowed us to visualize LTB₄ distribution and secretion in a variety of in vitro and in vivo experimental setups.

Our current understanding of chemotaxis is largely based on tissue culture experiments where cells migrate towards exogenously applied gradients of chemoattractants[2, 37]. Models can estimate the spatiotemporal distribution of chemoattractants, however, tools such as GEM-LTB₄ will be required to precisely map the diffusion and degradation of chemoattractants around migrating cells[11]. Methods to determine accurate parameters of diffusion coefficients or dissipation rates of chemoattractants in tissues have been limited, however, they are fundamental to build reliable mathematical models for describing cell migration. Measurements of exogenously applied chemoattractants in

live tissues, such as we showed here for LTB₄ in zebrafish, will be the basis to determine these parameters.

The power of measuring LTB₄ release on the single cell level will also aid our understanding of neutrophil heterogeneity[38]. Our results on bone marrow-derived neutrophils indeed indicate that the LTB₄-producing capacity is variable among these cells, which would not have been possible to determine from bulk ELISA measurements. How heterogeneity contributes to emergent neutrophil behaviors such as swarming, is a question that can be answered in the future with GEM-LTB₄. Using the sensor to assess the spatiotemporal production of LTB₄ could also add to our understanding of how intermediates during transcellular LTB₄ biosynthesis can be transported between neutrophils[15].

As the GEM-LTB₄ sensor is based on the high-affinity BLT1 receptor, high levels of its expression may affect endogenous LTB₄ availability. While we have demonstrated that its affinity is in the physiological range and overexpression of the sensor does not affect neutrophil wound recruitment during tissue damage, we also measured a shifted dose-response to exogenously applied LTB₄. As it has been proposed with other GPCR-based sensors such as dLight or GRAB_DA sensors, this buffering potential should be taken into consideration when end users interpret experimental results[39,40]. A further consideration that should be kept in mind with any fluorescent protein-based sensor such as GEM-LTB₄ is their pH sensitivity. While

alterations of extracellular pH have a minimal effect on GPCR-based neurotransmitter biosensors as shown before[24, 41], changes in intracellular pH of the cells that the sensors are expressed in could alter the measured fluorescence signal[23]. Using adequate controls such as the ligand-insensitive control version of the sensors should allow to rule out pH-related confounding effects.

Just as how GPCR-based biosensors became a key in recent years to understand the spatiotemporal coding in neuronal circuits[22–24], we expect fluorescent biosensors beyond GEM-LTB₄ to expand the horizons of immuno-imaging. We anticipate that our study will prime future developments of further GPCR-based sensors in the inflammation biology field to facilitate the long-sought live visualization of chemoattractants such as IL-8, C5a, fMLP or chemokines such as CCL19, 20 or 21[1, 37]. Furthermore, as it has been demonstrated for a number of GPCR-based sensors, it is possible to exchange the fluorophore to red-shifted variants[39, 40]. Future engineering efforts of new sensors should thus allow multiplex imaging to monitor the release of several chemoattractants at the same time.

## Methods

### Ethical Statement

All animal experiments were done with the approval of the Institutional Animal Care and Use Committee (IACUC) of Semmelweis University. All experimental procedures were approved by the Hungarian National Food Chain Safety Office (Permit Number: PE/EA/1027-7/2019).

### Cell lines

HEK293A cells were obtained from ThermoFisher (CAT#: R70507) and maintained in DMEM supplemented with 10% fetal bovine serum, 50 U/ml penicillin, and 50 μg/ml streptomycin in a 5% humidified $CO_2$ incubator at 37 °C.

For establishing stable HEK293A cell lines expressing our sensors, cells were co-transfected (see Methods Transfection) in a 1:1 ratio with *SB100x Sleeping Beauty* transposase and the SB transposon plasmids encoding the sensors. Cells were selected for at least two weeks using 2 μg/ml puromycin, and positive cells were then isolated by flow cytometry on a BD FACSAria™ III, using 488 and 561 nm excitation and 530/30 and 670/30 nm emission wavelengths. These cells were then maintained in complete DMEM as described above, supplemented with 0.5 μg/ml of puromycin.

### Zebrafish

Wild-type (AB) and Casper[42] strains were used for experimentation and the generation of transgenic lines. Experiments were performed on 3-4 days post-fertilization (dpf) larvae. Larval zebrafish do not have sex differentiation before 1-month post fertilization[43].

To generate our in-house transgenic lines, fresh fertilized embryos (30–60 min post-fertilized) were microinjected with transgenesis plasmids as described[44]. Positive embryos were selected according to their cardiac marker expression, and raised until sexual maturity to identify founder fish and establish F1 generations. We used the QF2/QUAS system[45], which in brief consists of a target promoter expressing the transcription activator QF2 and the QUAS enhancer (QF2 DNA binding site) introduced upstream of a target protein. QF2 and QUAS lines are generated separately, and when they are crossed together, the protein of interest is expressed in target cells.

The zebrafish lines used for experimentation include: *Tg(krt4:QF2)*, *Tg(krt19:QF2)*, *Tg(QUAS:PM-mKate2-P2A-GEM-LTB₄)*, *Tg(QUAS:PM-mKate2-P2A-GEM-LTB₄mut)*, *Tg(LysC:GCaMP7s-NES-P2A-mKate2-NES)* and *Tg(mpx:GFP)i114*[46].

After spawning and microinjection, zebrafish larvae were kept in E3 medium (5 mM NaCl, 0.17 mM KCl, 0.33 mM CaCl₂, and 0.33 mM MgSO₄) at 28 °C for 5-6 days before getting transferred to the main system. Adult fish were maintained as described[47], at 28 °C on a 14/10 h light/dark cycle.

### Plasmid construction

Human Leukotriene B4 receptor 1 (BLT1, Ensembl: ENSG00000213903) was cloned from complementary DNA derived from human peripheral blood mononuclear cells, and subcloned into the pEGFP-N1 (Clontech) vector backbone between XhoI and HindIII sites using standard molecular biology procedures. To create the GEM-LTB₄ prototypes, we introduced circularly permuted EGFP (cpEGFP) into the third intracellular loop of BLT1 between R212 and F213 using combinations of long (LSSLE) and short (GG) N- and C-terminal linkers. In a second round of screening, various lengths of the N-terminal LSSLI and C-terminal NHDQL linkers were combined. To facilitate the creation of sensor variants with different linkers, we introduced silent mutations resulting in unique restriction sites of PstI and SalI on the N- and C-termini of cpEGFP, respectively, outside of the linker sequences. To generate GEM-LTB₄mut, we introduced the R156A single amino acid mutation[26] into the BLT1 coding sequence of the final GEM-LTB₄ sensor.

To enhance the plasma membrane localization of the sensors, the mouse IgK leader sequence (METDTLLLWVLLLWVPGSTGD) was inserted upstream of the coding region. To coexpress a membrane-targeted red fluorescent protein beside GEM-LTB₄, we fused it N-terminally through a viral self-cleaving P2A peptide[48] with PM-mKate2, which uses the membrane localization sequence MGCVCSSNPENNNN, derived from the Lck protein.

To establish stable human embryonal kidney (HEK293A) cell lines, the sensors were subcloned into the "Sleeping Beauty" (SB) transposon plasmid allowing for puromycin-based selection[49, 50].

To create transgenic zebrafish lines, we relied on the Tol2kit system[44] combined with the QF2/QUAS system[51] (see "Zebrafish" paragraph) to express our sensors. To create plasmids for transgenesis, DNA fragments encoding PM-mKate2-P2A-GEM-LTB₄ and PM-mKate2-P2A-GEM-LTB₄mut were first subcloned into the pME backbone as entry clones, and recombined with the QUAS enhancer and SV40 polyadenylation sequence into the pDestTol2CR vector backbone. This backbone contains minimal tol2 elements and a cardiac promoter expressing the red fluorescent mKate2 marker for selection. To express QF2 in epithelial cells, we used the krt4[52] and krtt1c19e[53] (referred to as krt19) promoters driving expression in the suprabasal and basal epithelial cells, respectively, from a pDestTol2CG2 backbone harboring a cardiac green selection marker. The p5E-krt4, p5e-krtt1c19e, p5e-QUAS, p5e-lysC and pME-QF2 plasmids were kind gifts from Philipp Niethammer. The Tol2kit system was also used to create a transgenic line expressing the GCaMP7s (Addgene 104463) calcium sensor in the neutrophils through the *lysC* promoter[54]. Expression was restricted to the cytoplasm through a C-terminal nuclear export signal (NES), and mKate2-NES was also fused to GCaMP7s through a P2A peptide.

### Cell transfection

For transient DNA expression, HEK293A cells were seeded and transfected with lipofectamine 2000 (ThermoFisher), according to manufacturer's guidelines. In brief, we used 1 μl of lipofectamine with 500 ng of DNA for 2.5 cm2 of culture surface area. These numbers were then scaled up or down according to the surface area. Cells were then further incubated for 16–24 h before experiments.

To determine GEM-LTB₄ spectra, cells were electroporated using the Neon Transfection System (ThermoFisher), according to manufacturer's guidelines. In brief, we used 2 pulses at 1005 V for 35 ms on $1 \times 10^6$ cells in suspension. Cells were then further incubated for 16–24 h before experiments.

### Spinning disk confocal microscopy

All imaging was performed on an inverted Nikon Eclipse Ti2 microscope with a motorized piezo stage, perfect focus system and 488 nm and 561 nm laser lines for the Yokogawa CSU-W1 spinning disk scan head combined with two back illuminated Photometrics Prime BSI

scientific CMOS cameras for detection. Green and red fluorescence were recorded using 525/50 nm and 600/30 nm emission filters, respectively.

Images were recorded using the NIS Elements AR 5.4 software with a dimension of 1024 × 1024 pixels, unless otherwise stated, at 2 × 2 binning and bit depth of 16 bits. For acquisition, we used 40x Apo-Lambda/NA1.15 water-dipping and 20x ApoLambda/NA0.95 water-dipping objectives.

To determine the $\tau_{on}$ value of GEM-LTB$_4$, images were recorded with a dimension of 128 × 128 pixels with a frequency of 50 frames per second.

### Fluorescence imaging and treatment of cultured cells
HEK293A, PM-mKate2-P2A-GEM-LTB$_4$- or PM-mKate2-P2A-GEM-LTB$_4$mut-expressing stable HEK293A cells were seeded on poly-D-lysine-coated µ-slide 8-well plates (Ibidi) at a density of 30,000 cells/well. After 16–24 h, cells were transfected with plasmid DNA of interest and further incubated for 16–24 h, while stable sensor-expressing cells were directly incubated for 36–48 h. Before imaging, growth medium was replaced with transparent extracellular media (EC) consisting of: 3.1 mM KCl, 133.2 mM NaCl, 0.5 mM KH$_2$PO$_4$, 0.5 mM MgSO$_4$, 5 mM Na-Hepes, 2 mM NaHCO$_3$, 1.2 mM CaCl$_2$ and 2.5 mM glucose.

During cell experimentation, unless otherwise stated, a 2-3 min baseline was recorded before adding different ligands onto the cells. All eicosanoid ligands were obtained from Cayman Chemicals and include: LTB$_4$ (CAT#: 20110; CAS: 71160-24-2), Arachidonic Acid (CAT#: 90010; CAS: 506-32-1), 5-oxoETE (CAT#: 34250; CAS: 106154-18-1), 5(S)-HETE (CAT#: 34230; CAS: 70608-72-9), 12(S)-HETE (CAT#: 34570; CAS: 54397-83-0) and 20-Hydroxy LTB$_4$ (CAT#: 20190; CAS: 79516-82-8). Apart from the dose-response assay, all ligands were used at a final concentration (FC) of 100 nM. Normalized intensity changes were then measured and calculated (see below).

For establishing a dose-response curve, increasing LTB$_4$ concentrations were used, either a single concentration ranging from 1 pM to 1 µM or sequential concentrations of 1, 10 and 100 nM, given at 3 min intervals.

To inhibit GEM-LTB$_4$, we used the competitive antagonists of BLT1, CP-105,696 (Sigma-Aldrich, CAT#: PZ0363; CAS: 158081-99-3) and BIIL260 hydrochloride (Sigma-Aldrich, CAT#: SML2286; CAS: 204974-93-6). LTB$_4$ was added at a FC of 100 nM onto the cells after a 2 min baseline, then the inhibitors were applied either directly or perfused onto the cells using a custom-made perfusion system.

Ionophores Ionomycin (Cayman Chemical, CAT#: 10004974; CAS: 56092-81-0) and A23187 (Sigma-Aldrich, CAT#: C9275; CAS: 52665-69-7) were directly added onto the cells at a FC of 10 and 1 µM, respectively.

### Spectral scan
HEK293A cells were transfected with GEM-LTB$_4$. Before experimentation, cells were lifted and 1 × 10$^6$ cells were re-suspended in 100 µl EC media and either stimulated with 100 nM of LTB$_4$ or not. Fluorescence was measured using a CLARIOstar (BMG Labtech) plate reader. The excitation scan was performed by exciting from 340 to 520 nm with a step width of 2 nm, and collecting emission at 560/20 nm. The emission scan was determined by excitation at 470/20 nm and collecting emission from 490 to 650 nm with a step width of 2 nm.

### pH sensitivity assay
To assess the effect of pH on GEM-LTB$_4$ basal fluorescence intensity, PM-mKate2-P2A-GEM-LTB$_4$-expressing stable HEK293A cells were seeded as previously described and pre-incubated for 10 min in intracellular media (IC): 125 mM KCl, 20 mM NaCl, 0.5 mM MgCl$_2$, 0.2 mM K-EGTA and 20 mM buffering agent (see below) at pH 7.4, supplemented with nigericin (5 µg/ml) and monensin (5 µM). After

recording a 2 min baseline the pH of the media was changed by swapping to a different pH-adjusted IC media also supplemented with nigericin/monensin. The pH values of the IC media were set with the following buffering agents: MES for pH 6.2–6.6, MOPS for pH 7.0, HEPES for pH 7.4–7.8 or TRIS for pH 8.2–8.6.

To measure the effect of pH on GEM-LTB$_4$ responsiveness to LTB$_4$, cells were pre-incubated for 10 min in the different pH-adjusted IC media, supplemented with nigericin/monensin. After recording a 2 min baseline, LTB$_4$ (100 nM) was added to the cells for another 2 min.

### Isolation and stimulation of mouse neutrophils
Murine bone marrow cells were flushed from femurs and tibias using Ca$^{2+}$/Mg$^{2+}$ free HBSS and Phenol Red (Capricorn Scientific, CAT#: HBSS-2A) supplemented with 20 mM Hepes (Sigma-Aldrich, CAT#: H0887, CAS :7365-45-9). After centrifugation (5 min, 500 RCF) of the bone marrow, red blood cells were lysed with 5 ml 0.2% NaCl solution for 40 seconds, then the reaction was stopped with additional 5 ml 1.6% NaCl. To remove any remaining bone pieces the cells were strained through a 70 µm cell strainer (Corning), then centrifuged for 5 min at 1500 rpm and resuspended in 5 ml HBSS. The leukocytes were then loaded on top of 5 ml 62.5% Percoll (Sigma-Aldrich, CAT#: GE17-0891-02) and density gradient centrifugation was performed at 1300 RCF for 30 min at room temperature (RT) as previously described[55]. Finally, neutrophils were re-suspended in EC media and kept at room temperature until use.

To record endogenous LTB$_4$ release from activated neutrophils, after establishing a baseline of 2 min, 2 × 10$^6$ neutrophils/cm$^2$ were added on the top of the sensor-expressing stable HEK293A cells and immediately stimulated with 2 µM fMLP (Sigma-Aldrich, CAT#: F3506; CAS: 59880-97-6), or plain EC media. The experiment was recorded for an additional 28 min, in a total volume of 300 µl. Exogenous LTB$_4$ was then directly added into the wells 5 min before the end of the experiments.

To quantify LTB$_4$ levels from activated neutrophils, 2 × 10$^6$ neutrophils were stimulated with 2 µM fMLP or EC media, and incubated at RT for 28 min in 300 µl. An LTB$_4$ ELISA assay (ThermoFischer Scientific) was performed using the supernatant of the cells according to the manufacturer's instructions.

### Fluorescence microscopy and treatment of transgenic zebrafish
In vivo experiments were conducted on 3-4 days post fertilization (dpf) old larvae. Before experiments, larvae were anesthetized using 0.2 mg/ml Tricaine (Sigma-Aldrich, CAT#: 10521; CAS: 886-86-2) in isotonic E3 (standard E3 prepared with the additional 140 mM NaCl). Larvae were maintained and measured in isotonic solution to prevent early leukocyte recruitment[29] and wound closure[30], thereby allowing better penetration of exogenously applied substances. If needed, larvae were wounded (ventral nick-wound or tail fin amputation) using a 4 mm carbon steel needle blade micro knife (Fine Science Tools). Afterwards, unless otherwise stated, larvae were mounted in isotonic E3-based 1% low melting agarose (Gold biotechnology).

To test GEM-LTB$_4$ in vivo, exogenous LTB$_4$ was directly added onto intact or wounded larvae at a FC of 1 µM after recording a 5 min baseline. To measure endogenous LTB$_4$ production, larvae were wounded on the ventral tail fin and incubated for 90 min in isotonic E3 medium supplemented with arachidonic acid (FC = 20 µM) to attract the leukocytes to the wound[29,56]. E3 or 20 µM zileuton (Sigma-Aldrich, CAT#: 1724656; CAS: 111406-87-2), a 5-lipoxygenase inhibitor, was added 40 min before the end of the incubation. The larvae were then mounted in hypotonic (standard E3 embryo medium) 1% low melting agarose. After a 3 min baseline, A23187 (FC = 100 µM) was directly added onto the larvae to activate the cPLA$_2$ and 5-lipoxygenase enzymes. Endogenous LTB$_4$ production was also measured in untreated larvae which were wounded, mounted and imaged in normal E3 embryo medium.

In neutrophil migration assays, exogenous LTB$_4$ was directly added to wounded *Tg(mpx:GFP)i114* larvae after a 5 min baseline measurement. Alternatively, for migration assays without specific labeling of distinct leukocyte populations, intact or wounded larvae exposed to different exogenous concentrations of LTB$_4$ or normal E3 embryo medium were imaged by brightfield microscopy.

## Computational analysis and quantification

All image analysis was processed using in-house programming pipelines in Python. Prior to analysis, the background intensity was subtracted automatically using the SMO software[57] and images were registered using pyStackReg[58].

In experiments on HEK293A cells, GEM-LTB$_4$- and GEM-LTB$_4$mut-expressing cells were segmented in the red channel (mKate2) using the Cellpose software[59] and the generated masks were tracked using a software developed by Löffler et al. [60]. If needed, full tracked-masks were transformed to only include cell membrane masks. These masks were then used to extract the data, giving the average intensity of each channel (green and red) for each cell at each time point.

The extracted data were processed using the pandas Python library. Given that mKate2 (red) and GEM-LTB$_4$ (green) are expressed in a fixed ratio[48] and both are located in the same cell compartment (membrane), a green/red intensity ratio was calculated as a normalized signal. To specifically assess the effect of intracellular pH on GEM-LTB$_4$ fluorescence, we only utilized the green channel to quantify the measurements of the pH sensitivity assay.

To express all intensities to a relative baseline of 0, the formula $(F(t)-F_0)/F_0$, named as $\Delta F/F_0$, was applied on the data, where $F(t)$ is the intensity at a given time point and $F_0$ the average intensity of the baseline. In case of the cellular calcium assay, data were normalized between baseline and the ionomycin-induced maximal response values. Therefore, the following formula $(F(t)-F_0)/(F_{max}-F_0)$ was used, where $F_{max}$ is the average intensity after ionomycin stimulation.

To quantify endogenous LTB$_4$ secretion from murine neutrophils using sensor-expressing stable HEK293A lines, besides calculating pixelwise $\Delta F/F_0$ values as described above, a threshold value was applied on the red channel of the image to remove the background and create an overall mask of the cells. This mask was then subdivided into region of interests (ROIs) of $32 \times 32$ pixels in order to capture local changes in $\Delta F/F_0$ values. The mean intensities of these ROIs were then normalized in the same way as above. ROIs that display a minimum of 50% change in $\Delta F/F_0$ after fMLP stimulation compared to baseline were labelled as positive ROIs. The summed area of all positive ROIs was then compared to the overall area (size of the cell mask) to express the positive ROIs as coverage percentage.

To create a radial profile plot expressing the intensity distribution as a function of distance from a center point, the radial profile calculator or the diplib library (https://diplib.org/) was used.

To quantify relative plasma membrane localization of different GEM-LTB$_4$ prototypes and β-arrestin2, after generating individual masks for each cell, the masks were separately dilated and eroded and a differential mask was calculated to yield a ring-shaped mask covering the plasma membrane. The eroded mask was considered as the cytoplasm mask. Relative membrane localization was calculated as $F_{membrane}/F_{cytoplasm}$ or $F_{membrane}/F_{cell}$ after background subtraction.

For titration curves, the EC$_{50}$ value was obtained by fitting the data on a four-parameter log-logistic function.

To determine $\tau_{on}$ value of GEM-LTB$_4$, raw fluorescence intensity data from individual measurements of high-speed acquisition were smoothed by using a 500 msec wide raised cosine Hanning window. Intensity values were then normalized between average baseline (0%) and LTB$_4$ stimulated (100%) maximum fluorescence values. For each measurement, a horizontal curve was fitted onto the baseline and the maximal values and a rising linear curve on the middle subtriple of the dataset. The latter curve was used to determine the $\tau_{on}$ values, the time required to reach 50% of the maximal fluorescence increase. To correct for the msec differences between LTB$_4$ stimulation times among the experimental samples, a final normalization step was performed along the time axis of the data by setting the starting point of the rising curves (i.e., time of LTB$_4$ stimulation) to 0 msec. Finally, a normalized average activation curve was calculated by applying Hanning-smoothing on the combined baseline, the maximal horizontal and the rising linear curves determined above.

For in vivo experiments, images were pre-processed similarly as described above. Briefly, a threshold value was applied on the red channel of the image to remove the background and create an overall mask of the expressing cells. A "wound" mask was manually drawn to determine the edges of the wound as a reference. By using the distance transform function from the mahotas library[61], a distance gradient map was generated from the wound mask in order to determine intensity values as a function of their distance from the wound margin. Averaged intensities of each gradient layer (i.e. each distance layer from the wound) were binned (by $2 \times 2$ pixels) and green/red ratios were calculated. Finally, $\Delta F/F_0$ ratios were calculated based on baseline $F_0$ values.

For neutrophil migration quantification, cells expressing GFP were tracked and analyzed as described above. As for leukocyte migration quantification, cells were manually tracked using the MTrackJ plugin from ImageJ/Fiji[62]. To determine their position relative to the wound, a "wound" mask was also used for each time point. Leukocyte trajectories were analyzed in-depth as described before[63] yielding the parameters of average velocity (v), average path length (l), path linearity (Dp) and wound directionality (Dw).

## Statistics and reproducibility

No statistical methods were used to predetermine the sample sizes. In analyses of GEM-LTB$_4$ and GEM-LTB$_4$mut expressing HEK293A cells, data points were only excluded when the expression levels in the individual cells were so low, that the basal fluorescence of the sensor did not reach the manually adjusted intensity threshold (F sensor <11 over background). All attempts at replication were successful. The repeat (n) times are labeled in the corresponding figure legends. The work does not involve participant groups, therefore neither randomization nor blinding were used for the study.

Normalized and fully processed data were plotted using the seaborn library[64]. Montage images and videos were assembled using ImageJ/Fiji.

Statistical tests were calculated in Python using the Pingouin library[65]. For pairwise analysis, statistical significance was determined by a two-tailed unpaired Student's *t*-test with Welch's correction. For multiple comparison of independent conditions, one-way ANOVA was used with Fisher's LSD (Least Significant Difference) correction, or with Dunnett's correction, if a control condition was used. For multiple comparison of the same population over time, one-way repeated measure ANOVA was used with Bonferroni correction. All measurements are expressed as mean ± SEM, with sample size, number of replicates and *P* values indicated in figure legends.

## Reporting summary

Further information on research design is available in the Nature Portfolio Reporting Summary linked to this article.

# Data availability

Raw and source data along with codes for analysis have been uploaded to https://github.com/EnyediLab/ImageAnalysis_pipeline and are also available on Zenodo[66]. While source data are available for Fig. 3c, d and Fig. 4d, raw data are only available upon request, due to the large size of the files (~20 GB each). To request the raw data, please contact the corresponding author (enyedi.balazs@med.semmelweis-univ.hu). Requests will be fulfilled within 2 weeks. Source data are provided with this paper.

The main plasmids generated in this study have been deposited to Addgene: pSB-CMV-MCS-Puro GEM-LTB$_4$ (202641), pSB-CMV-MCS-Puro GEM-LTB$_4$mut (202642), pSB-CMV-MCS-puro PM-mKate2-P2A-GEM-LTB$_4$ (202643) and pSB-CMV-MCS-puro PM-mKate2-P2A-GEM-LTB$_4$mut (202644). Other plasmids and original material are available from the lead contact upon request.

## Code availability

All original codes have been deposited at https://github.com/EnyediLab/ImageAnalysis_pipeline and are also publicly available on Zenodo[66].

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

## Acknowledgements

The authors would like to thank P. Niethammer for his thoughtful comments on the manuscript, and P. Enyedi for helpful discussions along the way. We thank Zs. Ladányi and K. Kiss for assistance with experiments, Á. Marinkás for technical support, T. Németh and E. Káposztás for help with mouse neutrophil isolation and Gy. Várady for support on cell sorting. Research was supported by a "Lendület" grant from the Hungarian Academy of Sciences (LP2018-13/2018) and funding from the EU's Horizon 2020 research and innovation program under grant agreement No. 739593 (to B.E.). The work was also financed by the Thematic Excellence Program 2021 Health Subprogram (MOLORKIV) of the Ministry for Innovation and Technology in Hungary under project no. TKP2021-EGA-24 from the National Research, Development and Innovation Fund (to B.E.).

## Author contributions

B.E. conceived the project and designed the experiments. Sz.X.T., B.V. and B.T.R. performed the experiments. Sz.X.T., A.T., L.F. created zebrafish lines, B.T.R., F.G.D. and Sz.X.T. developed computational tools and wrote computer code to analyze the data. B.E., Sz.X.T. and B.T.R. participated in writing the manuscript.

## Funding

## Competing interests

The authors declare no competing interests.
