## [Peer Review File · Nature Communications]

REVIEWER COMMENTS

Reviewer #1 (Remarks to the Author):

The authors show a novel approach to solving a prominent problem in the immuno-imaging field. This research shows a significant advance in applied genetics as well as providing new insights into existing signalling pathways using both in vitro and in vivo models. The authors have generated a transgenic system for reporting the presence of the lipid mediator LTB4. They show extensive optimisation to generate this reporter, then characterise its specificity and sensitivity. They then go on to show the use of this reporter in vivo in a widely-used zebrafish tailfin injury system.

The manuscript makes a significant and convincing contribution to the field, especially for research regarding cell signalling dynamics. I have no major issues.

Minor points:

The authors should be cautious in their use of “neutrophils” and “leukocytes” in the last results paragraph as well as figure 4, not only did this introduce some confusion, there is no direct proof that it is indeed neutrophils migrating towards the wound in the zebrafish model. Further methods such as a sudan black staining or the use of a neutrophil specific transgenic line would help in clarifying this point.

The discussion mentions how the GEM-LTB4 sensor could expand the horizons of immuno-imaging, but could benefit further from more concrete examples.

Allele codes should be given for all transgenic zebrafish lines. Zebrafish nomenclature requires italicisation of the line names.

Where transgenic lines were obtained from external sources, this should be indicated and the allele code given.

Details of the code deposition in GitHub and plasmid deposition in Addgene are missing.

It is an accepted convention that larvae are positioned with head to the left in imaging studies.

Reviewer #2 (Remarks to the Author):

The manuscript of Tamás and coworkers describes a genetically encoded fluorescent probe for leukotriene B4 visualization in situ and in vivo. The authors describe the design of the probe, characterize it in cell cultures and in zebrafish in vivo. Clearly, the probe has several features that make it useful. First, it has ~2-fold amplitude of response, which is enough for the majority of settings. Second, it has a great selectivity towards LTB4 and responds to low nanomolar concentrations of the ligand. Its performance in vivo is very good, although seemingly the model that the authors use for in vivo justification produces very low amounts of LTB4 in the absence of exogenously applied calcium ionophore. Although I'm positive about the probe and the paper, I have several questions, which I believe will help to strengthen the manuscript and clarify some issues.

1. It is not clear how bright the probe is. Although it is not possible to express it in e.coli and purify, it would help to compare brightness with, let say, PM targeted EGFP in cell culture using FACS or microscopy.
2. It is not clear what a pH dependency of the probe is. Calibration in cell culture with nigericin/monensin and a range of buffers would help.
3. I completely miss the point why the authors use mKate together with GEM-LTB4 probe to achieve ratiometricity of the latter. The probe is intrinsically ratiometric, it has two excitation peaks, like many cpFP based green probes have, and it would be more suitable to excite sequentially 500 and 400 peaks with green emission. Adding RFP via p2a sequence makes the DNA cassette larger and adds another protein to express.
4. LTB4 is a chemoattractant for neutrophils themselves. However, the authors never express the probe in neutrophils or HL-60 cells to visualize if, for example, they see higher LTB4 at the leading edge of migrating cells.
5. It would be great to understand if there are some shortcomings of the probe. E.g., to what extent the probe expression changes the endogenous LTB4 gradients? How neutrophil migration speed to the wound changes in TG vs wt animals?

Reviewer #3 (Remarks to the Author):

This manuscript reports the development and validation of a new GPCR-based fluorescent sensor for leukotriene B4 (LTB4). The authors followed the design of recently engineered transmitter sensors, and generated the sensor by comparing properties of a small number of variants containing mutations in a few key positions. Without using a large scale mutagenesis and screening, this approach generated a LTB4 sensor with decent sensitivity, specificity, fluorescence response up on ligand binding, and minimal non-specific effects. The sensor appeared to allow the real-time visualization of both exogenously applied and endogenously produced LTB4 gradients.

REVIEWER COMMENTS

Reviewer #1 (Remarks to the Author):

The authors show a novel approach to solving a prominent problem in the immuno-imaging field. This research shows a significant advance in applied genetics as well as providing new insights into existing signalling pathways using both in vitro and in vivo models. The authors have generated a transgenic system for reporting the presence of the lipid mediator LTB₄. They show extensive optimisation to generate this reporter, then characterise its specificity and sensitivity. They then go on to show the use of this reporter in vivo in a widely-used zebrafish tailfin injury system.

The manuscript makes a significant and convincing contribution to the field, especially for research regarding cell signalling dynamics. I have no major issues.

- **We are pleased that the Reviewer considers our work to make a significant contribution to the field and that only minor questions have been raised. These have all been addressed in the revised manuscript as detailed below.**

Minor points:

The authors should be cautious in their use of “neutrophils” and “leukocytes” in the last results paragraph as well as figure 4, not only did this introduce some confusion, there is no direct proof that it is indeed neutrophils migrating towards the wound in the zebrafish model. Further methods such as a sudan black staining or the use of a neutrophil specific transgenic line would help in clarifying this point.

- **We appreciate the Reviewer for pointing out the confusing use of “neutrophils” vs. “leukocytes” throughout the original manuscript. For the sake of consistency, in the revised version we only use “neutrophils” if *Tg(mpx:GFP)i114* or *Tg(LysC:GCaMP7s-NES-P2A-mKate2-NES)* lines were used to indeed identify this cell population. In every other experiment where brightfield imaging was used to locate migrating cells in the tissue, we now use the more general “leukocyte” term.**

In the experimental setup discussed in the last paragraph, we did not yet have a chance to fluorescently label neutrophils along with the GEM-sensors, as it would have required us to cross three lines (*Tg(krt19:QF2)* x *Tg(QUAS:PM-mKate2-P2A-GEM-LTB₄)* x some other neutrophil-specific line). Nevertheless, to confirm that neutrophils indeed arrive to the wound site in the experimental setup shown in Figure 4, we now include a time-lapse video (Supplementary video 5a) of a *Tg(LysC:GCaMP7s-NES-P2A-mKate2-NES)* transgenic line, which was treated exactly as the GEM-LTB₄ expressing larvae. Besides expressing a neutrophil-specific mKate2 marker, this line also expresses GCaMP7s in the neutrophils. This allowed us to confirm that the ionophore A23189 is indeed penetrating into the fish to increase the cytoplasmic [Ca²⁺] in the neutrophils. Nevertheless, as we are not directly proving that neutrophils are the source of the measured LTB₄ signal, we have changed the wording in this paragraph as well.

The discussion mentions how the GEM-LTB₄ sensor could expand the horizons of immuno-imaging, but could benefit further from more concrete examples.

- **We thank the Reviewer for asking us to provide a longer discussion on the potential future use of GEM-LTB₄ and other GPCR-based probes. In this paragraph of the original paper, we were mainly eluding to future developments of sensors beyond GEM-LTB₄, that could detect molecules such as IL-8, fMLP, C5a and further chemoattractants and chemokines. We are describing this more clearly in the revised version of the manuscript.**

Allele codes should be given for all transgenic zebrafish lines. Zebrafish nomenclature requires italicisation of the line names. Where transgenic lines were obtained from external sources, this should be indicated and the allele code given.

- **Allele codes and transgenic zebrafish line nomenclature has now been updated according to Zfin guidelines. We have also initiated to deposit the data on the transgenic lines created in this study at Zfin.**

Details of the code deposition in GitHub and plasmid deposition in Addgene are missing.

- **We have made our image analysis code available under: https://github.com/EnyediLab/ImageAnalysis_pipeline The core plasmids encoding the GEM-LTB₄ sensor along with its mutant version have been deposited at Addgene under accession numbers 202641-202644.**

It is an accepted convention that larvae are positioned with head to the left in imaging studies.

- **We thank the reviewer for pointing this out - we have accordingly modified all images and illustrations in Fig.3 and Fig.4.**

Reviewer #2 (Remarks to the Author):

The manuscript of Tamás and coworkers describes a genetically encoded fluorescent probe for leukotriene B₄ visualization in situ and in vivo. The authors describe the design of the probe, characterize it in cell cultures and in zebrafish in vivo. Clearly, the probe has several features that make it useful. First, it has ~2-fold amplitude of response, which is enough for the majority of settings. Second, it has a great selectivity towards LTB₄ and responds to low nanomolar concentrations of the ligand. Its performance in vivo is very good, although seemingly the model that the authors use for in vivo justification produces very low amounts of LTB₄ in the absence of exogenously applied calcium ionophore. Although I'm positive about the probe and the paper, I have several questions, which I believe will help to strengthen the manuscript and clarify some issues.

- **We appreciate the Reviewer's positive comments and thoughtful suggestions on the manuscript. By addressing the questions and providing an even more detailed characterization of the sensor, we believe to have made the design and interpretation of future experiments with GEM-LTB₄ easier for end users in the research community.**

1. It is not clear how bright the probe is. Although it is not possible to express it in e.coli and purify, It would help to compare brightness with, let say, PM targeted EGFP in cell culture using FACS or microscopy.

- **We compared the fluorescence of GEM-LTB₄ with EGFP-tagged BLT1R, which shows a similar PM-localization as the sensor (Supplementary Fig. 4) and found that the brightness of the sensor in LTB₄-bound state was about 7-fold lower than that of the tagged receptor (Supplementary Fig. 1f).**

2. It is not clear what a pH dependency of the probe is. Calibration in cell culture with nigericin/monensin and a range of buffers would help.

- **Fluorescent proteins are intrinsically pH-sensitive. Therefore, we greatly appreciate the reviewer for requesting a detailed description of the pH dependency of GEM-LTB₄ using nigericin/monensin treated cells. In fact, none of the currently published GPCR-based GRAB- or dLight-sensors were characterized as thoroughly as we have now done this for GEM-LTB₄. Publications showing minimal pH sensitivity of the cpEGFP-based probes measure fluorescence in intact, non-permeabilized cells, using a set of extracellular buffers set to different pH values -**

e.g. (Duffet et al., 2022; Ino et al., 2022). Not surprisingly, the fluorescence values of the sensors remain relatively stable, as cells employ various mechanisms to maintain their intracellular pH within a narrow range despite fluctuations in extracellular pH.

In the revised version of the manuscript we are now providing a state-of-the art pH-calibration of the sensor using nigericin/monensin treated cells, in which intracellular pH values indeed follow the pH of the surrounding buffer. We created buffers at various pH values using MES, MOPS, HEPES or TRIS with a compositions resembling the intracellular milieu: high K^+ (~145mM) and low Na^+ (~20mM) levels. Two experimental setups were carried out, which are now shown as Supplementary Fig. 3.

1, The pH 7.4 buffer of the nigericin/monensin treated cells was swapped using a perfusion system to a range of pH values from 6.2-8.6. As reported for all GFP-derived fluorescent proteins, acidic media decreases, whereas alkaline media increases the fluorescence of the probe. The relative change in baseline fluorescence was in the range of -65% to +40% at the two measured extreme pH values, 6.2 and 8.6, respectively, as shown in the new Supplementary Fig 3.a.

2, Nigericin/monensin treated cells were kept in a set of various intracellular buffers described above (pH 6.2-8.6), and were stimulated with 100 nM LTB_4 to determine the change in dF/F_0 values (Supplementary Fig 3.b). The relative change in fluorescence of GEM- LTB_4 was ~ 45% ($\Delta F/F_0 = 46 \pm 0.2\%$) in the treated cells kept in a 7.4 pH intracellular buffer. This is lower compared to the $\Delta F/F_0 = 103 \pm 1\%$ signal increase of the sensor which can be measured in intact cells, and is possibly due to the altered ionic composition and membrane potential resulting from the nigericin and monensin treatment. Nevertheless, the LTB_4 -induced relative $\Delta F/F_0$ was stable in a range of 7.4-8.6, indicating that alkalization of the intracellular milieu of the cells does not alter the reactivity of the sensor to LTB_4 . Decreasing the pH below 7.4 gradually decreased the reactivity of the sensor, however, GEM- LTB_4 was still reactive at all of the measured pH values (Supplementary Fig 3.b).

Together, these experiments clearly show that caution should be taken with GEM- LTB_4 , just as with any other fluorescent biosensor: potential alterations of the pH in the cells expressing the sensor should be accounted for. Proper controls such as using the mutant version of the sensor or inhibitors (e.g. zileuton in Fig 4, or experiments with mutant controls) will confirm if the change in the measured fluorescence is due to the measured ligand or an unspecific change of the intracellular pH.

To make this point clear in the manuscript, we have now devoted a paragraph in the revised version on precautions that should be taken when using GEM and other GPCR-bases sensors.

3. I completely miss the point why the authors use mKate together with GEM- LTB_4 probe to achieve ratiometricity of the latter. The probe is intrinsically ratiometric, it has two excitation peaks, like many cpFP based green probes have, and it would be more suitable to excite sequentially 500 and 400 peaks with green emission. Adding RFP via p2a sequence makes the DNA cassette larger and adds another protein to express.

- Indeed the reviewer is completely right that GEM- LTB_4 is intrinsically ratiometric and 500/400 nm ratiometric measurements would have also been possible throughout our experiments. Nevertheless, using intensity-change based probes which are dim in the unstimulated form is sometimes challenging *in vivo* (especially in mice or and zebrafish): both identifying the animals which express the probe and also the reliable segmentation and tracking of individual cells is often problematic if a strong and constant fluorescence imaging channel is not available (Duffet et al., 2022). All of our image-processing pipelines use the Cellpose algorithm to automatically identify individual cells, and we used the P2A-fused mKate2 channel for the sake of creating these cellular masks. To stay consistent with the imaging setup throughout the experiments shown in the manuscript, we imaged this red channel during all of our experiments, and we did not want to use an extra channel (405) to avoid extra phototoxicity and bleaching. To make sure that any change in (488 excited) GEM-

LTB₄ fluorescence is not due to focal drifts, we used the mKate2 channel to calculate a “pseudo-ratiometric” value.

In future experiments where a red channel would be used for other localization- or intensity-based readouts, we would also recommend the use of a 488/405 nm excitation based ratiometric imaging if necessary.

4. LTB₄ is a chemoattractant for neutrophils themselves. However, the authors never express the probe in neutrophils or HL-60 cells to visualize if, for example, they see higher LTB₄ at the leading edge of migrating cells.

- We thank the reviewer for this suggestion and agree that future experiments and projects using the GEM-LTB₄ sensor will allow the scientific community to address questions such as proposed one. To show that GEM-LTB₄ indeed works in neutrophils or neutrophil-like cells, we have created a stable GEM-LTB₄-expressing HL-60 cell line and show in the figure below that it can be used to detect LTB₄.

Revision Fig.1 | a, GEM-LTB₄ confocal fluorescence and corresponding ΔF/F₀ ratio images in HL-60 cells before and after 100 nM LTB₄ stimulation. Scale bar, 20 μm **b**, GEM-LTB₄ ΔF/F₀ fluorescence responses in HL-60 cells before and after 100 nM LTB₄ stimulation. (n=50, +/- SE).

However, the primary goal of our paper has been to demonstrate the power and use of GEM-LTB₄. Tackling specific biological questions such as further details in the concept of transcellular LTB₄ production, how and which neutrophils LTB₄ is released from during swarming or which part of a migrating cell LTB₄ is secreted from are complex questions well worth the investigation. We believe that future studies with a broader scope to better understand specific aspects of neutrophil biology should deal with these questions. Addressing such questions in our study would take the focus away from our methodological paper. Moreover, we hope to share this tool as soon as possible with the research community and allow fellow scientists to pursue their own interests with it.

5. It would be great to understand if there are some shortcomings of the probe. E.g., to what extent the probe expression changes the endogenous LTB₄ gradients? How neutrophil migration speed to the wound changes in TG vs wt animals?

- To make this point clear in the manuscript, we have now devoted a paragraph in the revised version of the manuscript on possible challenges with GEM and other GPCR-bases sensors including potential ligand buffering effects and also that caution that should be taken regarding alterations of intracellular pH-values. Regarding the ligand buffering potential of GEM-LTB₄, based on new experiments that we have performed and now show in Supplementary Fig. 6, we can confirm what others (Labouesse and Patriarchi, 2021; Sun et al.,

2020), have proposed and seen as a potential drawback of GPCR-based sensors: their high expression levels in transgenic animals can shift the dose-response towards the measured ligand. We now show that neutrophil dissemination triggered by 30 nM LTB₄ is significantly lower in GEM-LTB₄ expressing zebrafish compared to wt or GEM-LTB₄mut transgenic animals. This difference is not present at a higher dose of LTB₄ (100 nM). At the same time, neutrophil migration towards wounds is not significantly different in GEM-LTB₄ Tg vs. wt animals: the number of neutrophils recruited to the wound within an hour post wounding, their migration speed (v), average path length (l), wound directionality (Dw) and path persistency (Dp) were only different in a range that was not statistically significant (Supplementary Fig. 6b).

Reviewer #3 (Remarks to the Author):

This manuscript reports the development and validation of a new GPCR-based fluorescent sensor for leukotriene B4 (LTB₄). The authors followed the design of recently engineered transmitter sensors, and generated the sensor by comparing properties of a small number of variants containing mutations in a few key positions. Without using a large scale mutagenesis and screening, this approach generated a LTB₄ sensor with decent sensitivity, specificity, fluorescence response up on ligand binding, and minimal non-specific effects. The sensor appeared to allow the real-time visualization of both exogenously applied and endogenously produced LTB₄ gradients.

- We thank Reviewer's the positive comments on the manuscript.

References:

- Duffet L, Kosar S, Panniello M, Viberti B, Bracey E, Zych AD, Radoux-Mergault A, Zhou X, Dernic J, Ravotto L, Tsai Y-C, Figueiredo M, Tyagarajan SK, Weber B, Stoeber M, Gogolla N, Schmidt MH, Adamantidis AR, Fellin T, Burdakov D, Patriarchi T. 2022. A genetically encoded sensor for in vivo imaging of orexin neuropeptides. *Nat Methods* **19**:231–241. doi:10.1038/s41592-021-01390-2
- Ino D, Tanaka Y, Hibino H, Nishiyama M. 2022. A fluorescent sensor for real-time measurement of extracellular oxytocin dynamics in the brain. *Nat Methods* **19**:1286–1294. doi:10.1038/s41592-022-01597-x
- Labouesse MA, Patriarchi T. 2021. A versatile GPCR toolkit to track in vivo neuromodulation: not a one-size-fits-all sensor. *Neuropsychopharmacol* **46**:2043–2047. doi:10.1038/s41386-021-00982-y
- Sun F, Zhou J, Dai B, Qian T, Zeng J, Li X, Zhuo Y, Zhang Y, Wang Y, Qian C, Tan K, Feng J, Dong H, Lin D, Cui G, Li Y. 2020. Next-generation GRAB sensors for monitoring dopaminergic activity in vivo. *Nat Methods* **17**:1156–1166. doi:10.1038/s41592-020-00981-9

REVIEWERS' COMMENTS

Reviewer #1 (Remarks to the Author):

The authors have adequately addressed our concerns.

Reviewer #2 (Remarks to the Author):

All my comments were correctly addressed. I think the ms can be accepted for publication.

REVIEWERS' COMMENTS

Reviewer #1 (Remarks to the Author):

The authors have adequately addressed our concerns.

- **We are pleased to have successfully addressed all the issues raised, and we would like to express our gratitude for the valuable feedback of the reviewer. Their input has played a crucial role in enhancing both the quality and clarity of our paper.**

Reviewer #2 (Remarks to the Author):

All my comments were correctly addressed. I think the ms can be accepted for publication.

- **We are delighted to receive your positive feedback, and are grateful for your invaluable contribution throughout the review process, which has played a significant role in refining and improving our work. Thank you for your recommendation to accept the paper.**